# Physical characteristics of soil-biodegradable and nonbiodegradable plastic mulches impact conidial splash dispersal of *Botrytis cinerea*

**Xuechun Wang[1], Chakradhar Mattupalli[2], Gary Chastagner[3], Lydia Tymon[3], Zixuan Wu[4], Sunghwan Jung[4], Hang Liu[5], Lisa Wasko DeVetter**[1]*

1 Department of Horticulture, Washington State University, Northwestern Washington Research and Extension Center, Mount Vernon, Washington, United States of America, 2 Department of Plant Pathology, Washington State University, Northwestern Washington Research and Extension Center, Mount Vernon, Washington, United States of America, 3 Department of Plant Pathology, Washington State University, Puyallup Research and Extension Center, Puyallup, Washington, United States of America, 4 Department of Biological and Environmental Engineering, Cornell University, Ithaca, New York, United States of America, 5 Department of Apparel, Merchandising, Design and Textiles, Washington State University, Pullman, Washington, United States of America

* lisa.devetter@wsu.edu

**Data Availability Statement:** All relevant data are within the paper and its Supporting Information files.

## Abstract

*Botrytis cinerea* causes gray mold disease of strawberry (*Fragaria* ×*ananassa*) and is a globally important pathogen that causes fruit rot both in the field and after harvest. Commercial strawberry production involves the use of plastic mulches made from non-degradable polyethylene (PE), with weedmat made from woven PE and soil-biodegradable plastic mulch (BDM) as emerging mulch technologies that may enhance sustainable production. Little is known regarding how these plastic mulches impact splash dispersal of *B. cinerea* conidia. The objective of this study was to investigate splash dispersal dynamics of *B. cinerea* when exposed to various plastic mulch surfaces. Mulch surface physical characteristics and conidial splash dispersal patterns were evaluated for the three mulches. Micrographs revealed different surface characteristics that have the potential to influence splash dispersal: PE had a flat, smooth surface, whereas weedmat had large ridges and BDM had an embossed surface. Both PE mulch and BDM were impermeable to water whereas weedmat was semi-permeable. Results generated using an enclosed rain simulator system showed that as the horizontal distance from the inoculum source increased, the number of splash dispersed *B. cinerea* conidia captured per plate decreased for all mulch treatments. More than 50% and approximately 80% of the total number of dispersed conidia were found on plates 10 and 16 cm away from the inoculum source across all treatments, respectively. A significant correlation between the total and germinated conidia on plates across all mulch treatments was detected ($P<0.01$). Irrespective of distance from the inoculum source, embossed BDM facilitated higher total and germinated splashed conidia ($P<0.001$) compared to PE mulch and weedmat ($P = 0.43$ and $P = 0.23$, respectively), indicating BDM's or embossed film's potential for enhancing *B. cinerea* inoculum availability in strawberry

**Funding:** This project was funded by the Washington State Department of Agriculture Specialty Crop Block Grant program (#K2863). Additional funding was provided by the Specialty Crops Research Initiative Award 2022-51181-38325 from the USDA National Institute of Food and Agriculture. Any opinions, findings, conclusions, or recommendations expressed in this publication are those of the author(s) and do not necessarily reflect the view of the U.S. Department of Agriculture. Corresponding author, L.W.D., received the awards. The funders had no role in study design, data collection and analysis, decision to publish, or preparation of the manuscript.

**Competing interests:** The authors have declared that no competing interests exist.

production under plasticulture. However, differences in conidial concentrations observed among treatments were low and may not be pathologically relevant.

## Introduction

Fungi have evolved different mechanisms to aid dispersal of their propagules [1]. The main dispersal mechanisms of fungal plant pathogens are wind and rain splash [1–6]. Understanding dispersal mechanisms is important for limiting fungal propagule movement and reducing the spread of global diseases that threaten food security and plant survival [7]. Fungal plant pathogens belonging to phylum Ascomycota are capable of producing asexual spores known as conidia and these conidia are regarded as the major dispersal unit of many important plant pathogens [8]. Conidia can be dispersed over short distances (e.g., a few centimeters) by rain-generated splash droplets or over long distances (e.g., several kilometers) by wind [9]. Infected plant tissues are a source of conidial inoculum and conidia incorporated into water droplets can be transported to neighboring plants by rain or irrigation splash events [10]. Rain splash dispersal can become critical in the development of fungal disease epidemics when a large number of conidia are transported to neighboring plants [11,12]. Previous research has demonstrated there is variability in conidial dispersal numbers through rain splash [10,13,14] and this variability is attributed to multiple factors including the surfaces that splashed droplets are interacting with [9,15,16]. However, the overall role of surface characteristics on splash dispersal of plant pathogens is understudied and in need of further investigation to inform disease risk and associated management.

Gray mold is an important fungal disease caused by *Botrytis cinerea* that severely impacts yields in commercial strawberry (*Fragaria ×ananassa*) production systems [2,17,18]. The pathogen is found worldwide [3], produces abundant inoculum in the form of conidia [4], and is considered a primary pathogen responsible for significant yield losses of harvested strawberry across the world [17]. Yield losses due to this pathogen can exceed 80% in the absence of fungicides and under favorable environmental conditions [18]. The pathogen is currently considered hemibiotrophic [19] and can infect a variety of tissues including flowers and fruits (Fig 1). Ripe and green strawberry fruits are both susceptible to this pathogen in the field [20,21]. *Botrytis* is also regarded as a post-harvest pathogen because the disease can develop during transportation and storage [22,23]. Rain splash dispersal may cause severe secondary gray mold infections through the dispersal of conidia from infected flowers, buds, or fruits to susceptible tissues [11,24,25], especially in high humidity (>80%) environments or when rainfall occurs preharvest [26,27].

Commercial strawberry production is often done under plasticulture using non-degradable polyethylene (PE) mulch (Fig 2) [28,29]. PE mulch utilization promotes plant growth and increases yield primarily through weed suppression and optimization of soil temperatures [30–32]. Weedmat (a semi-permeable, woven, PE- or polypropylene-based geotextile) is an alternative to PE that is preferred by some farmers due to its durability [33] and potential to be re-used over multiple seasons. Soil-biodegradable plastic mulch (BDM) is perceived as a more sustainable alternative to PE and weedmat because it is designed to biodegrade in soils and consequently reduce plastic waste generation [34,35].

Ground covers including mulches can vary in surface characteristics and impact splash dispersal of plant pathogens. For example, splash dispersal patterns of *Colletotrichum acutatum* (cause of anthracnose) and *Phytophthora cactorum* (cause of leather rot on strawberry fruits and Phytophthora crown and root rots) were highly related to ground cover physical characteristics [6,36,37]. Random roughness was determined to be one of the key physical features of

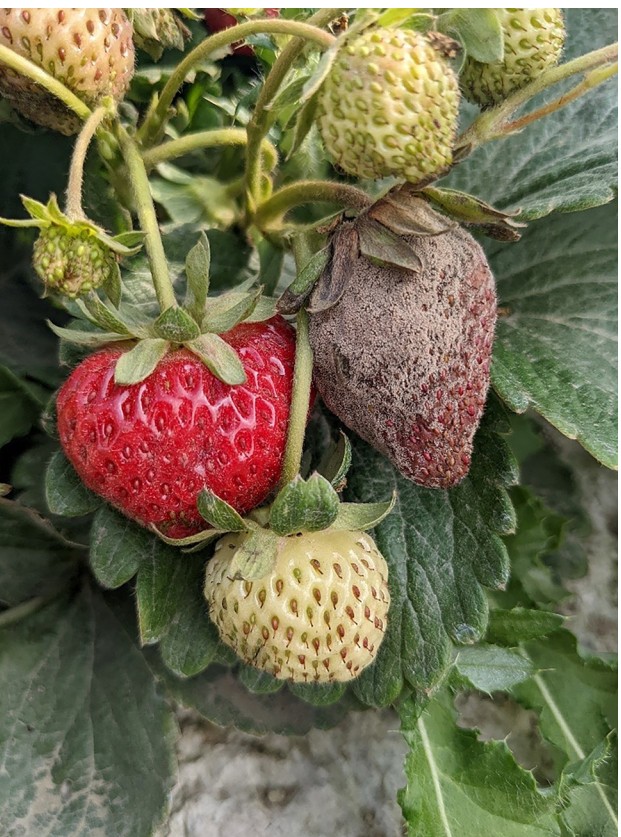

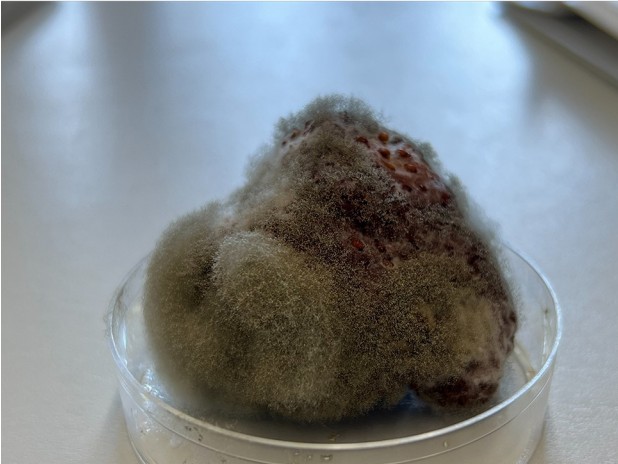

**Fig 1.** Infected strawberry fruit covered with gray mold (*Botrytis cinerea*) (A). Sporulating *B. cinerea* after inoculation of strawberry fruits with a SDHI (succinate dehydrogenase inhibitor; FRAC-7) fungicide-resistant isolate (B). Sporulation demonstrates the pathogenicity of the *B. cinerea* isolate used in the experiment. Photos by DeVetter (A) and Baral (B).

ground cover surface microtopography that influence splash dispersal [38] with decreasing roughness increasing the magnitude of splash distance [36,39]. For example, smooth PE mulch generated more conidial splash dispersal of *C. acutatum* compared to a straw mulch [36]. However, the impact of groundcover random roughness is not consistent as a buckwheat

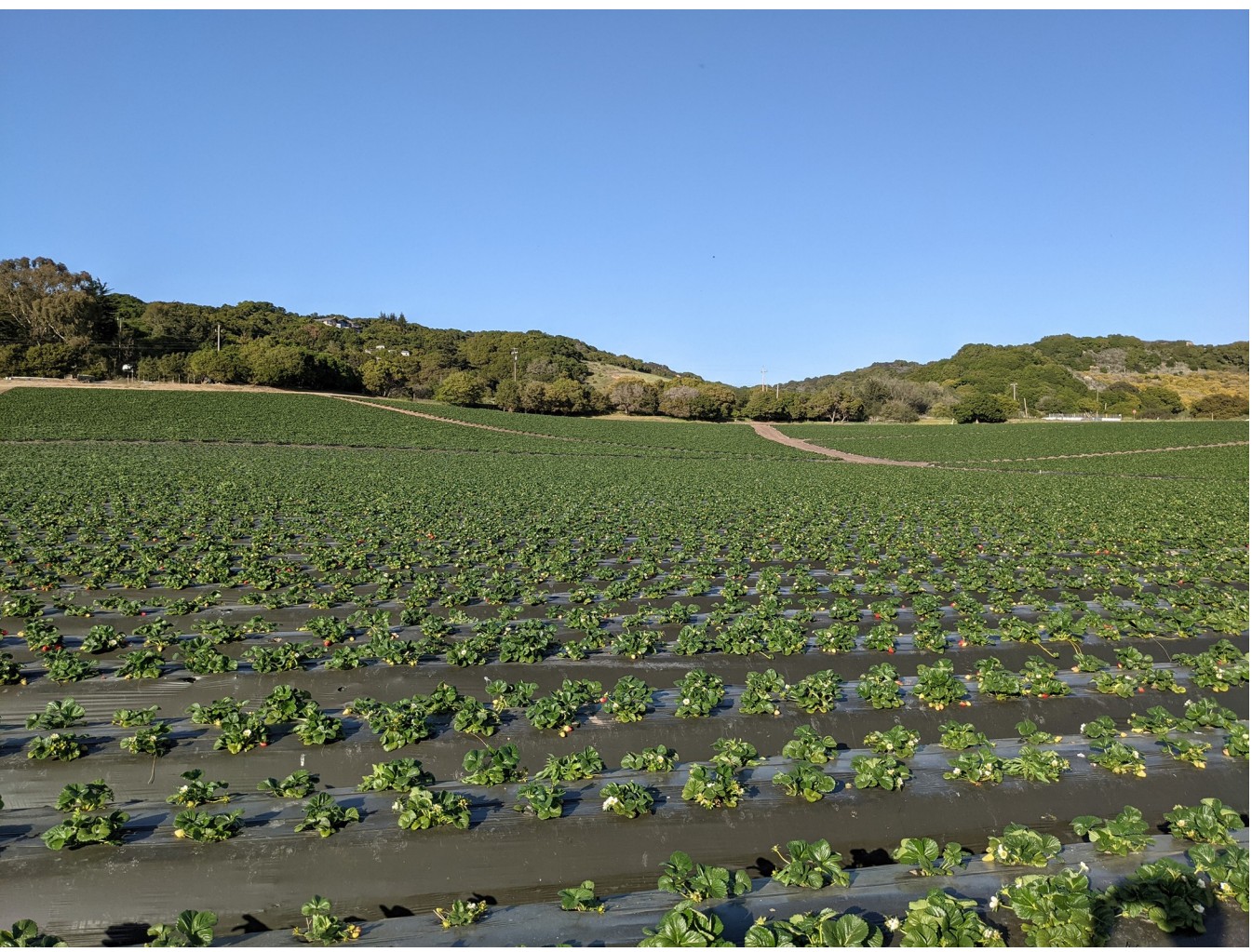

**Fig 2. Commercial strawberry is typically grown using plastic polyethylene (PE) mulches, a practiced referred to as plasticulture.** Photo by DeVetter.

(*Fagopyrum esculentum*) husk mulch increased gray mold incidence in strawberry compared to other mulches including black plastic, wood chips, and straw [40]. Other ground cover characteristics such as permeability and surface adhesion of water may impact the dynamics of pathogen splash dispersal. However, there are no recent studies on splash dispersal from mulch surfaces and a systematic understanding of *B. cinerea* conidial dispersal via rain splash in strawberry productions systems under plasticulture is lacking.

The objective of this study was to investigate splash dispersal dynamics of *B. cinerea* conidia with three plastic mulch surfaces: PE mulch, weedmat, and BDM. Specific sub-objectives were to: 1) Evaluate mulch surface physical characteristics that could impact splash dispersal dynamics and 2) Determine whether different plastic mulches affect conidial splash dispersal patterns of *B. cinerea*.

## Materials and methods

### Mulch surface physical characteristics

New, unweathered PE, weedmat, and embossed BDM mulches included in this study (Table 1) were the same products that had been used in a previous mulch experiment [41].

**Table 1. List of mulch treatments with thickness and feedstock ingredients provided by mulch manufacturers.**

| Mulch treatment | Thickness/density | Converter/manufacturer | Key feedstock ingredients |
|---|---|---|---|
| Polyethylene | 25.5 μm | Filmtech, LLC., Stanley, WI, USA | Polyethylene |
| Soil-biodegradable mulch, embossed | 25.5 μm | Organix Solutions, Bloomington, MN, USA | PLA+PBAT[a] |
| Weedmat | 85 g/m² | Extenday, Union Gap, WA, USA | Woven polyethylene |

[a]PLA = polylactic acid; PBAT = polybutylene adipate-co-terephthalate.

Surface microtopographic features of mulch samples were assessed using a digital microscope (Keyence VHX-7000) in the Composite Materials and Engineering Center at Washington State University (WSU) in Pullman, Washington, USA. Mulch permeability was characterized to assess how material properties affect infiltration of liquid involved in splash dispersal and subsequent conidial dispersal outcomes. For each mulch treatment, a large plastic Petri dish (14 cm in diameter) was filled with greenhouse growing medium (Steuber Promix BX General Purpose, Premier Tech Horticulture, Quakertown, PA, USA) and covered with one of the mulch treatments. A 20 cm-long square piece of mulch was secured tightly around the dish by taping the mulch to the bottom of the Petri dish and adjusting the mulch surface tension to 0.74 N at the center using a tension meter (Chatillon 516 Series linear push/pull scale, Ametek, Berwyn, PA, USA), which is comparable to mulch tension measurements collected in the field (S1 Table). A silicone ring (6 cm diameter and 1 cm high) was placed on the mulch surface in the center of the Petri dish and sealed to the mulch using a white, 100% silicone sealant (Gorilla Glue Company; Berlin, Germany). The sealant was allowed to dry completely (~30 min) before 10 ml of water was gently poured into the area within the silicone ring. The volume of remaining water within the silicone ring was measured after 2, 4, 8, 16, and 32 minutes at 25°C using a 10 ml-graduated cylinder. Permeability measurements were repeated three times as technical replicates for each mulch treatment using new mulch for each repetition.

## Characterization of splash dynamics

To better understand splash dynamics of water on PE mulch, weedmat, and BDM surfaces, a preliminary non-replicated experiment using a single droplet was conducted. A water nozzle, syringe pump (NE-1000, Pump Systems LLC., Dickinson, ND, USA), and needle attached to the nozzle were used to generate individual water droplets falling onto either PE mulch, weedmat, or BDM surfaces. A high-speed camera (NOVA S6, Photron USA, Inc., San Diego, CA, USA) with a backlit LED light source was used to record the splash dynamics initiated by a second drop falling onto the former drop placed on mulch surfaces. The number of resulting satellite droplets as well as qualitative data describing splash dynamics were recorded.

## Conidial splash dispersal of *B. cinerea*

This experiment was conducted in a fully enclosed screen house (15 m long, 6 m wide, 5.5 m tall with 270 × 770 μm mesh size; Gable Series 7500, U.S. Global Resources, Seattle, WA, USA) at the WSU Mount Vernon Northwestern Washington Research and Extension Center (WSU NWREC) in Mount Vernon, WA to minimize potential contamination. The ground of the screenhouse was covered with woven, polyethylene landscape fabric. Within the screenhouse an Eurmax Premium instant canopy tent with enclosed sidewalls (3 m long, 2.9 m wide, 2.3 m tall; Eurmax Canopy, EI Monte, CA, USA) was assembled to further minimize contamination potential and wind effects (Fig 3).

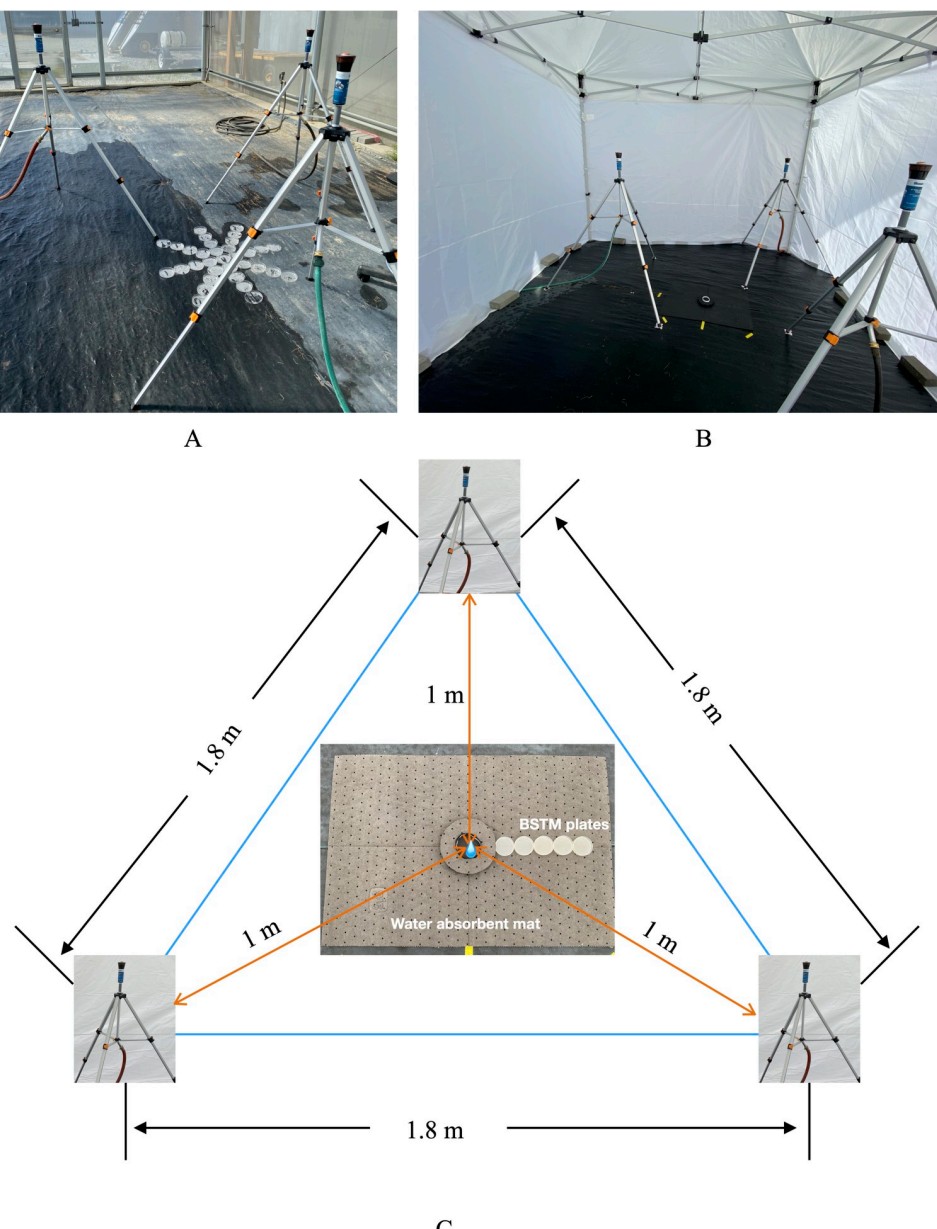

**Fig 3. Rain simulator system design for mimicking rain events for the splash dispersal experiment.** Three pulsating sprinklers were mounted on tripods and uniformity tests were conducted to adjust the arrangement of sprinklers (A). A canopy tent was established to enclose the experiment and block the potential for wind dispersal impacts (B). Three sprinkler tripods were arranged in a triangle shape at specific distances (C) and the mulch treatment was placed at the centroid of the triangle shape. At the center of the mulch treatment, a conidial suspension $(2 \pm 0.2 \times 10^6$ conidia/ml, symbolized as blue dot in the picture where three orange arrows point) was placed on the mulch treatment to serve as the inoculum source; five *Botrytis* spore trap media plates (BSTM) were placed adjacent to each other and arranged in a line. The mulch treatment and BSTM plates were placed on the water absorbent mat (C). Photo by Wang.

A rain simulator system was built within the canopy tent consisting of three metal pulsating sprinklers (Hunter MP Rotator® MP800, Hunter Industries, San Marcos, CA, USA) mounted on tripods (Melnor 65066-AMZ; Melnor Inc., Frederick County, VA, USA; Fig 3). There was one sprinkler head per tripod and individual sprinkler heads were arranged in an equilateral

triangle with each side measuring 1.8 m long between sprinkler heads (Fig 3C). Each sprinkler head consisted of a high-efficiency nozzle paired with a pop-up body (Hunter Pro-Spray PRS30, Hunter Industries, San Marcos, CA, USA) for pressure regulation to 207 KPa. A schedule 80 PVC nipple (Grainger Industrial Supply, Lake Forest, IL, USA) was used to connect the sprinkler head and irrigation pipes. Water was discharged from nozzles at a height of 1.5 m.

Uniformity tests were conducted prior to conidial dispersal experiments to position the nozzles, maximize head-to-head coverage, and optimize uniformity of water discharged from the rain simulator system (Fig 3A). Each uniformity test was performed for five minutes and repeated three times as technical replicates for each configuration until maximum uniformity was achieved. Water was collected into five empty Petri plates (10 cm diameter) placed in the central area where water was discharged by the rain simulator system. A total of 40 plates were used for each uniformity test with plates arranged in eight cardinal and ordinal directions (with 5 plates in each direction) radiating from the center (North, South, West, East, Northwest, Northeast, Southwest, and Southeast). Accumulated water was collected from each plate and measured at the end of each uniformity test. All three nozzles were adjusted to a 210˚ arc and a low flow rate (68 liter/hour). The central position of the sprinkler system was arranged so it was 1 m equidistant from the inoculum source (Fig 3B and 3C). The configuration of the rain simulator system used for subsequent conidial dispersal experiments achieved an 85–90% uniformity within 1.8 m (S2 Table), which included the area where the dispersal experiment was conducted. Additionally, multiple drops of water emitted from the rain simulator system were collected on water-sensitive paper. From this, the average size (i.e., diameter) of the drops was measured. Then 10, 20, 30, 40, 50, and 60ul of water were dropped individually on a separate piece of water-sensitive paper using a pipette. The size of the drops from the rain simulator system was compared to the drops containing a known volume of water to estimate average droplet volume from the rain simulator system. This approach allowed us to determine the average volume of a drop from the rain simulator, which was 50ul. Precipitation rate was also measured during uniformity tests and the pressure was adjusted to achieve a rate of 21 mm/ hour.

Following establishment of the rain simulator system, a conidial suspension was prepared using a pathogenic, SDHI (succinate dehydrogenase inhibitor; FRAC-7) fungicide-resistant isolate of *B. cinerea* to distinguish it from other airborne *Botrytis* spores during this experiment (Fig 1B). This isolate was characterized as having the P225F mutation which confers resistance to all SDHI active ingredients (Kozhar, personal communication). To prepare the conidial suspension, the isolate was cultured on full-strength potato dextrose agar amended with 100 μg/ ml salicylhydroxamic acid (SHAM), 32 μg/ml isofetamid, and 35 mg/ml chloramphenicol. Then *B. cinerea* culture plates were incubated at 22˚C in an incubator (Percival Intellus Ultra temperature controller, Percival Scientific, Inc., Perry, IA, USA) with 18 hours of white light (light intensity 320 lux) and 6 hours dark for 10 days. Sterile deionized (DI) water (20 mL) was added onto 10-day-old *B. cinerea* culture plates and a lab spatula was used to dislodge conidia into the water solution. The solution was then filtered through one layer of cheesecloth and transferred into 50 ml centrifuge tubes. The conidial suspension was vortexed for 5 minutes to distribute clumped conidia and then the conidial concentration was determined using a hemocytometer (Bright-line Model 1492, Hausser Scientific, Horsham, PA, USA). The final conidial concentration was adjusted to $2 \pm 0.2 \times 10^6$ conidia/ml. The conidial germination rate prior to running the splash dispersal experiment was nearly 100% (S3 Table).

Small Petri dishes (60 mm diameter) containing *Botrytis* spore trap media (BSTM) (modified from [42]) was used to collect the conidia splashed from the inoculum source. BSTM consisted of 2 g glucose, 20 g bacto™ agar, 0.1 g $NaNO_3$, 0.1 g $K_2HPO_4$, 0.2 g $MgSO_4 \cdot 7H_2O$ and 0.1 g KCl in one liter of DI water. The solution was autoclaved and cooled to 56˚C before

adding 2.5 g/l tannic acid and adjusting the pH to 4.5 with 1 mol/l NaOH. Next, 0.2 g/l chloramphenicol, 0.02 g/l pentachloronitrobenzene (PCNB), 0.02 g/l dithane, and 0.1 ml/l 12% fenarimol were added and the media was poured into Petri dishes.

Before running the rain simulator, a large piece of Pig® Absorbent Mat Pad (30 cm × 50 cm) was placed on the ground to limit secondary splash from outside of the primary inoculum source and each replicate was placed on the mat at a marked position within the center of the rain simulator system (Fig 3C). The absorbent mat was immediately replaced after each replicate. For each replicate, five of the BSTM plates were placed immediately adjacent to the Petri dish containing the inoculum and one of the three mulch treatments and arranged linearly in a southward direction (Fig 3C). This direction was based on uniformity results and maintained throughout the experiment for consistency. Each BSTM plate was immediately adjacent to the other, so the BSTM plate distance between each center of an adjacent BSTM plate was standardized at 6 cm. The center of the first BSTM plate was placed 10 cm away from the center of inoculum source and mulch treatment. Succeeding plates were placed at 16, 22, 28, and 34 cm from the center of the first plate. These distances from the inoculum source were assigned as dispersal distances (denoted as i, i = 10, 16, 22, 28, 34).

The experiment was repeated five times as technical replicates per mulch treatment with the construction of the mulch treatment done using the procedures outlined above for mulch permeability characterization. Additionally, after the silicone sealant dried, the annulus of the mulch surface was covered by a water absorbent mat (Pig® Absorbent Mat Pad, model MAT 203; New Pig Corporation, Tipton, PA) to reduce production of secondary splash droplets (Fig 3C). New mulch and absorbent mats were used for each replicate. The mulch surface and silicone band were then disinfected with 70% ethanol immediately before exposure to the rain simulator system. The *B. cinerea* conidial suspension (10 ml per replicate) was pipetted into the well created by the silicone ring and formed a pool on the treatment mulch. Immediately after the placement of BSTM plates and the addition of conidial suspension, the rain simulator system was initiated for two minutes with three people turning the irrigation on and off at the same time. BSTM plates were immediately collected and incubated at 22 ˚C for 18 hours with lids open in the same incubator listed above to dry the media surface and provide time for conidia to germinate. Prior to counting conidial germination, lids were placed back on the BSTM plates, and maintained at 1˚C for a maximum of 4 hours while conidial counts were being performed to minimize changes in spore germination during the counting process. Presence of a germ tube (length was greater than half the width of the conidia) growing out of a conidium was used to assign germination status. All conidia from BSTM plates were counted within 3 hours after removal from the incubator. Total dispersed conidia ($TC_i$) and the number of dispersed conidia that germinated after 18h ($GC_i$) were enumerated for each dispersal distance (i) from each BSTM plate using a compound microscope (Nikon Eclipse 50i, Nikon Inc., Melville, NY, USA) at 40X magnification.

## Data analysis

All quantitative data were analyzed using R Studio software (Version 1.4.1106, Rstudio PBC, Boston, MA, USA). Mulch permeability was graphed using ggplot2 package [43], while surface characteristics and splash dynamics were visually assessed. Findings from the non-replicated splash dynamics study were treated as qualitative observations and not analyzed due to the lack of replication. Conidial splash dispersal $TC_i$ data were first fit to an exponential model =

$$N = N_0 exp(-d/\tau) \tag{1}$$

with $N_0$ as the prefactor of the exponential model, which represents the number of conidia at

the impact location ($d = 0$). $\tau$ is the decay length and $N$ represents the number of conidia at distance $d$. Data are presented in their original units. Then two additional variables were calculated and added into further data analysis as follows:

The percentage (%) of $TC_i$ of total dispersed conidia at all distances [abbreviated as $T_i$byT] =

$$\frac{TC_i}{\sum_i TC_i} \times 100, \ i \in \{10, \ 16, \ 22, \ 28, \ 34\} \tag{2}$$

The percentage (%) of $GC_i$ of germinated conidia at all distances [abbreviated as $G_i$byG] =

$$\frac{GC_i}{\sum_i GC_i} \times 100, \ i \in \{10, \ 16, \ 22, \ 28, \ 34\} \tag{3}$$

Technical replicate was regarded as a random factor, while mulch treatment and dispersal distance were treated as fixed factors. Four variables ($TC_i$, $GC_i$, $T_i$byT, and $G_i$byG) were checked to ensure they met distribution assumptions and fitted into a generalized linear mixed-effects model. Normal distributions of those four variables were checked before depositing data into analysis using the his() function on residuals. According to different distribution patterns, $TC_i$ and $GC_i$ data were subjected to a Poisson distribution and log transformed, while $T_i$byT and $G_i$byG data were subjected to a normal distribution. An interaction between mulch treatment and dispersal distance was considered for these four variables. Multivariate Analysis of variance (MANOVA) was conducted using estimated marginal means (least-square means) with a post hoc Tukey–Kramer test for detection of treatment effects at a significance of $\alpha = 0.05$. Data were presented in their original units. Additionally, a Pearson product-moment correlation coefficient analysis was conducted between each of $TC_i$, $GC_i$, $T_i$byT, $G_i$byG and dispersal distance, between $TC_i$ and $GC_i$, and $T_i$byT and $G_i$byG. All data are available in Excel file S5.

## Results and discussion

Splash dynamics and dispersal of *B. cinerea* conidia differed due to mulch treatment and distance with treatment effects attributed to differences in the physical characteristics of the mulch surfaces. PE mulch, weedmat, and embossed BDM showed different surface topographic characteristics (Fig 4A–4C). PE mulch had a smooth, flat surface compared to the embossed BDM. Weedmat mulch had a rigid surface due to the woven fibers of PE. Furthermore, PE mulch and embossed BDM were impermeable to water and weedmat was semi-permeable based on laboratory observations (Fig 5). There was no water loss through infiltration for PE mulch and embossed BDM, whereas >94% of the water infiltrated through the weedmat after 32 mins. Mulch permeability in the screenhouse was not measured given there was additional water delivered via the rain simulator system and water displacement occurring from rain droplets impacting the inoculum pool. Although the woven construction of weedmat seemed to produce visibly similar surface characteristics as embossed BDM, the semi-permeability of weedmat led to infiltration of water containing the inoculum through the mulch and effectively reduced the volume of the conidial suspension.

Results from the rain simulator system showed the means of all $TC_i$ across the five distances for PE mulch, weedmat, and embossed BDM were 26.8, 31, and 47 conidia, respectively. All mulch treatments showed an inverse relationship in that as the horizontal distance from the inoculum source increased, the number of $TC_i$ and $GC_i$ decreased (Table 2). Inverse relationships were also observed between $T_i$byT, $G_i$byG and horizontal distance from the inoculum source (Table 2). Interestingly, the $TC_i$ and $GC_i$ differed due to mulch treatment ($P<0.001$ for

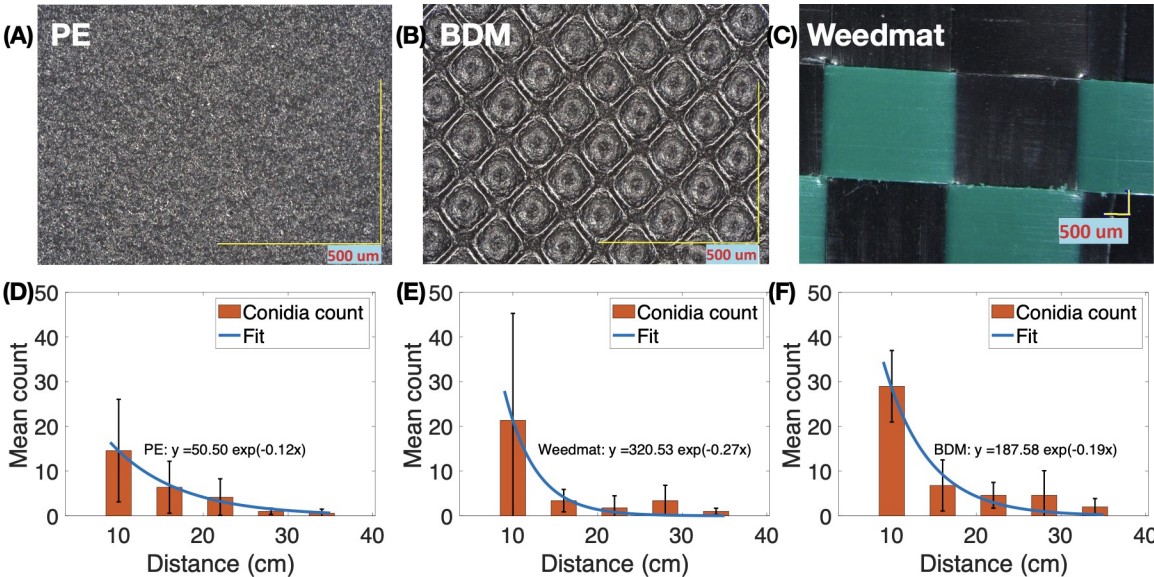

**Fig 4.** Surface microtopographic characteristics of polyethylene (PE; A), embossed soil-biodegradable plastic (BDM; B), and weedmat mulches (C). Scale bar denotes 500 μm. The number of dispersed *B. cinerea* conidia over distance is below (D-F). Blue lines are a model-fit using the function $N = N_0 exp(-d/\tau)$. Here $N_0$ is the prefactor of the exponential model, which represents the number of conidia at the impact location ($d = 0$). $\tau$ is the decay length and $N$ represents the number of conidia at distance $d$.

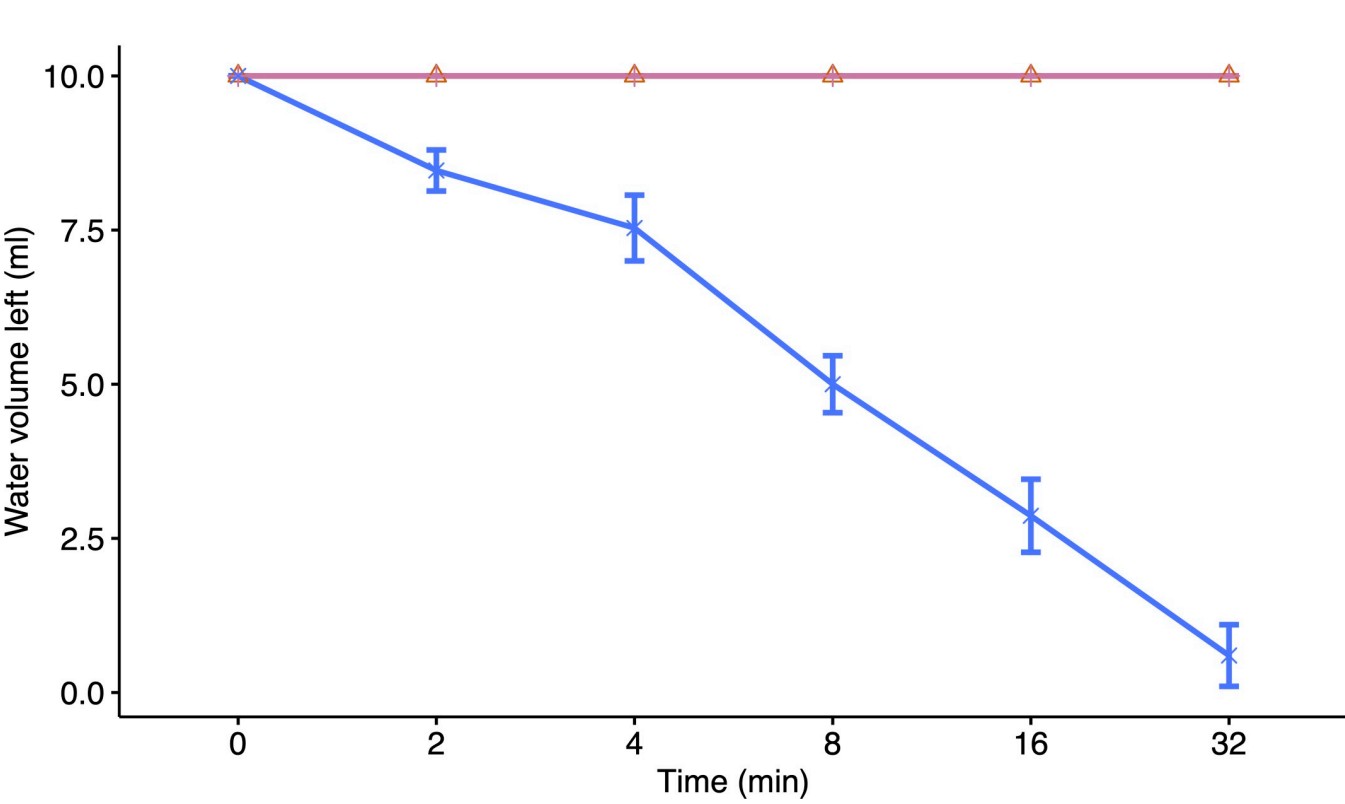

**Fig 5. Surface water retention of polyethylene (PE), embossed soil-biodegradable plastic (BDM), and weedmat mulches measured as the volume of remaining water on mulch surfaces after 2, 4, 8, 16, 32 minutes.**

**Table 2. Pearson product-moment correlation coefficient analysis between variables $TC_i$, $GC_i$, $T_ibyT$, $G_ibyG$ by dispersal distance and between $TC_i$ [a] by $GC_i$ [b] and $T_ibyT$[c] by $G_ibyG$[d].** Note i represents dispersal distance (cm) with i = 10, 16, 22, 28, 34.

| Variable A | Variable B | *P*-value | Coefficient value |
|---|---|---|---|
| $TC_i$[a] | Dispersal distance | <0.01 | -0.588 |
| $GC_i$[b] | Dispersal distance | <0.01 | -0.567 |
| $T_ibyT$[c] | Dispersal distance | <0.01 | -0.689 |
| $G_ibyG$[d] | Dispersal distance | <0.01 | -0.669 |
| $TC_i$ | $GC_i$ | <0.01 | 0.985 |
| $T_ibyT$ | $G_ibyG$ | <0.01 | 0.955 |

[a]Total dispersed *B. cinerea* conidia ($TC_i$).

[b]Number of dispersed conidia that germinated after 18h ($GC_i$).

[c]The percentage (%) of $TC_i$ of total dispersed conidia at all distances ($T_ibyT$).

[d]The percentage (%) of $GC_i$ of germinated conidia at all distances ($G_ibyG$).

**Table 3. Splash dispersal of *Botrytis cinerea* conidia on polyethylene (PE), embossed soil-biodegradable mulch (BDM), and weedmat by distance.** Values following the means are standard error.

| Mulch treatment | Dispersal distance (i)[a] | $TC_i$ [b] | $GC_i$ [c] | $T_ibyT$ (%)[d] | $G_ibyG$ (%)[e] |
|---|---|---|---|---|---|
| BDM | 10 | 29±3.6 A a[f] | 24±2.5 A a | 62±2.9 a[g] | 71±2.6 a[g] |
| Weedmat | 10 | 21±10.7 B a | 17±9.2 B a | 59±9.1 | 72±18.6 |
| PE | 10 | 15±5.1 C a | 12±4.5 B a | 46±15.3 | 57±17.3 |
| BDM | 16 | 9±2.5 A b | 5±1.8 A b | 18±3.3 b | 15±4.5 b |
| PE | 16 | 6±2.6 AB b | 4±2.4 B b | 31±15.6 | 31±18.2 |
| Weedmat | 16 | 3±1.1 B b | 2±1 B b | 21±9.8 | 21±19.7 |
| BDM | 22 | 5±1.3 Ab | 2±0.6 A bc | 10±2.5 bc | 5±1.8 b |
| PE | 22 | 4±1.8 AB b | 2±1.1 B bc | 15±7.0 | 12±7.0 |
| Weedmat | 22 | 2±1.2 B b | 1±0.8 B bc | 3±2.1 | 2±1.6 |
| BDM | 28 | 5±2.5 A b | 4±2.5 A b | 10±4.8 bc | 10±6.1 b |
| Weedmat | 28 | 3±1.5 A b | 2±1.5 B b | 11±3.0 | 5±3.1 |
| PE | 28 | 1±0.3 B c | 0 B b | 5±1.7 | 0 |
| BDM | 34 | 2±0.8 A b | 0±0.2 A c | 4±1.5 c | 2±1.0 c |
| Weedmat | 34 | 1±0.3 A b | 0 B c | 6±2.8 | 0 |
| PE | 34 | 1±0.4 A c | 0 B c | 3±2 | 0 |
| *P*-values | | | | | |
| Mulch treatment (m) | | <0.001 | <0.001 | 0.97 | 0.98 |
| Dispersal distance (i) | | <0.001 | <0.001 | <0.001 | <0.001 |
| m×i | | 0.01 | 0.20 | 0.64 | 0.88 |

[a] i represents dispersal distance (cm), i = 10, 16, 22, 28, 34.

[b] Total dispersed conidia ($TC_i$).

[c] Number of dispersed conidia that germinated after 18h ($GC_i$).

[d] The percentage (%) of $TC_i$ of total dispersed conidia at all distances ($T_ibyT$).

[e] The percentage (%) of $GC_i$ of germinated conidia at all distances ($G_ibyG$).

[f] Uppercase letters (A, B, C) indicate groupings of mulch treatment effects in either $TC_i$ and $GC_i$ at the same dispersal distance. Lowercase letters (a, b, c) indicate grouping of dispersal distance effects in $TC_i$, $GC_i$, $T_ibyT$, and $G_ibyG$ from the same mulch treatment. Values sharing same uppercase or lowercase letters indicate they were not statistically different (*P*>0.05).

[g]$T_ibyT$ and $G_ibyG$ did not have mulch treatment effects and the groupings for dispersal distance were calculated after averaging across mulch treatment.

$TC_i$ and $GC_i$,) and dispersal distance from the inoculum source ($P<0.001$) (Table 3). The $T_i$byT and $G_i$byG differed due to dispersal distance ($P<0.001$) but not due to mulch treatment ($P = 0.97–0.98$). There was an interaction between mulch treatment and dispersal distance ($P = 0.006$) for $TC_i$, but not for $GC_i$ ($P = 0.20$), $T_i$byT ($P = 0.64$), and $G_i$byG ($P = 0.88$). The $TC_i$ and $GC_i$ splashed from embossed BDM were the highest, followed by PE and weedmat mulches. PE and weedmat had similar $TC_i$ and $GC_i$ ($P = 0.43$ and $P = 0.23$ respectively). The dispersal distance at 10 cm and 34 cm had the highest and lowest $TC_i$ and $GC_i$, respectively, across all mulch treatments. $GC_{28}$ equaled to zero for PE mulch and $GC_{34}$ equaled to zero for PE and weedmat mulches. $T_{10}$byT and $G_{10}$byG was > 60% and $T_{10}$byT and $T_{16}$byT were approximately 80% across all mulch sources.

Differences in conidia dispersal distance were likely due to mulch surface topographic characteristics as the number of conidia over distance followed a simple exponential decay model (Fig 4D–4F). The measured $N_0$ [number of conidia at impact location ($d = 0$)] was 50.50 for PE mulch, 187.58 for embossed BDM, and 320.53 for weedmat. The decay lengths ($\tau$) were 8.33 cm for PE mulch, 5.26 cm for embossed BDM, and 3.70 cm for weedmat. This indicates that as mulch roughness increases, more conidia spread with shorter horizontal distance ($N_0$ increases and $\tau$ decreases). Physically, this may be observed as the rough surface of mulches create splashes more upward compared to a smooth surface.

Random roughness of groundcover surface microtopography has been determined to be a key physical feature that influences splash dispersal [36,38]. Preliminary observations from the non-replicated splash dynamics experiment showed that the second single drop falling onto an existing single drop on the weedmat and embossed BDM surfaces produced similar numbers of splashed satellite droplets (approximately 20–22), whereas PE produced fewer satellite droplets (approximately 15). Both PE and embossed BDM showed a well-connected splash crown upon secondary drop impact (Fig 6D and 6F). Furthermore, the rim of the crown ejected what appeared to be visually smaller splashed satellite droplets when interacting on the PE mulch surface than embossed BDM. Weedmat exhibited an irregular pattern of splashing (Fig 6F), and this was attributed to the large-scale roughness of the interwoven weedmat fibers. We also observed PE and embossed BDM surfaces produced a similar spherical adhesion to a single drop of water whereas the weedmat surface produced a flat shape (Fig 6A–6C), which may be due variations in mulch permeability (Fig 5) [44]. Although this was not a replicated trail, observations suggest that an irregular splashing pattern may produce a variable but sometimes

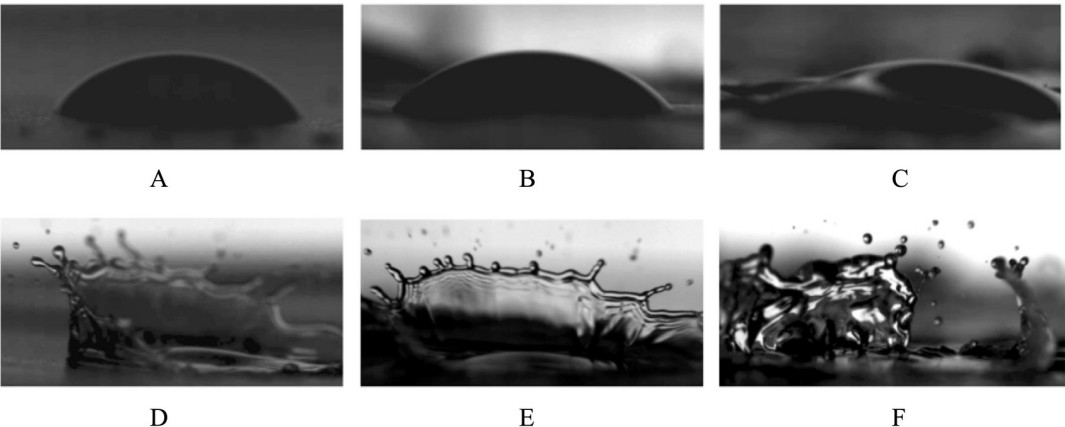

**Fig 6.** Splash dynamics of polyethylene (A and D), embossed soil-biodegradable plastic (B and E) and weedmat (C and F) mulches.

large number of splashed satellite droplets. Additionally, the embossed surface of BDM and interwoven fibers of weedmat lead to relatively upward splash patterns and a greater number of droplets than the PE mulch and this may increase the number of conidia dispersed in satellite droplets but quickly decay horizontally as described in the exponential model fit.

Regardless of mulch treatment, most *B. cinerea* conidia were dispersed within 10 cm from the inoculum source in this study. Conidia captured on media plates decreased greatly from 10 to 22 cm and were nearly absent at 34 cm from the inoculum source. A similar trend was observed for $GC_i$. It is also to be noted that there was a highly significant linear correlation between TCi and GCi, as well as between $T_i$byT and $G_i$byG (Table 2). Results from this study align well with previous research on plant pathogen spread by splash dispersal in rain-simulated systems. Previous work has demonstrated that the majority of splashed droplets are dispersed over short distances and larger droplets containing dozens to hundreds of conidia fall within 30 cm from inoculum sources [10,36,45–49]. Perryman et al. [10] also found larger splashed droplets containing an average of 308 conidia/droplet of *Pyllosticta citricarpa* (cause of Citrus black spot) were dispersed within 30 cm from the inoculum source while smaller splash droplets can be dispersed further and as far as 70 cm. Findings from our study are also in agreement with the general understanding that rain splash dispersal of fungal plant pathogens occurs at smaller scales (e.g., centimeters) compared to wind dispersal, which can disperse propagules several meters or even kilometers from an inoculum source [50–53].

The conidial density of *B. cinerea* necessary for infection and lesion development on strawberry flowers and fruits as well as the splash dispersal distance of water droplets with conidial concentrations needed for infection are currently unknown. Previous inoculation studies on grape (*Vitis vinifera*) berry surfaces showed an individual airborne conidium could infect berry tissue and lead to lesion development, but overall infection was governed by host resistance that varied between fresh and cold-stored fruits [54]. For strawberry, a *B. cinerea* concentration of $3 \times 10^3$ conidia/$m^3$ in air caused only 0.56% Botrytis fruit rot [55]. Studies demonstrating the density of *B. cinerea* needed in a liquid suspension to cause disease are limited. In lentil (*Lens culinaris*) seedlings, a *Botrytis* spp. suspension at $10^4$ spores/ml concentration caused a 20% chance of infection whereas a $10^2$ spores/ml concentration did not result in any symptom development [56]. The concentration of our suspension was $2 \pm 0.2 \times 10^6$ conidia/ml, which may be sufficient to cause disease. Hence, although our splash dispersal results showed a mulch treatment effect, whether the greater conidial numbers generated from embossed BDM cause more severe disease needs to be demonstrated using field-relevant conidial concentrations in suspension and is an area for future research. Decreasing random roughness (defined as the standard deviation of elevations over an area of interest from a base plane) [38] of ground covers, including mulches, has been previously reported to increase the magnitude of pathogen splash distance [6,24,57,58]. In this study, mulch random roughness was not measured, but visual assessment of mulch surface microtopography indicated PE to have lower mulch roughness than embossed BDM and weedmat due to their respective embossed and ridged surface features. PE can also be made with an embossed surface, so it is still an open question if similar responses will be observed between PE and BDM depending on whether or not they are embossed. Despite this variation in surface microtopography, all mulch treatments in the current study were associated with a decrease in the number of dispersed conidia with increasing horizontal dispersal distance. Furthermore, other factors such as the degree, uniformity, and scale of roughness may contribute to the magnitude of splash dispersal of pathogens.

It is to be noted that the assayed dispersal distances under our experimental conditions was designed on a small scale to reflect the range of typical plant spacings (25 to 30 cm) observed in commercial strawberry systems [23,59]. Our results suggest that wider within-row

strawberry plant spacing could reduce the spread of *B. cinerea* conidia transported through splash dispersal. Wider within-row plant spacing has been shown to reduce disease incidence of *Botrytis* in field-grown strawberry in Florida with 40% less disease at a wider spacing of 45.6 cm compared to 22.9 cm [60]. Other splash dispersal studies similarly found plant density can affect splash dispersal of *C. acutatum* in strawberry [61]. Although wider spacing can be beneficial for reducing plant pathogens spread by rain splash and reducing the period of free moisture on plants, there is a trade-off as this effectively reduces plant density and yield potential by having fewer plants per unit area.

Rain characteristics, such as rain density, duration, and droplet size are also known to impact pathogen dispersal distance [12,24,36,39,62]. Pathogens could furthermore have different splash dispersal distances due to different density and/or sizes of pathogen spores incorporated within a droplet [6,10,24,39]. However, studies on splash dispersal epidemiology of fungal plant pathogens are few and there are many unexplored aspects that remain understudied. It is also unclear if the feedstock ingredients of mulches contribute to differences in surface characteristics and how they interact with water, which could impact splash dispersal. BDMs are comprised of 75–95% biodegradable feedstock ingredients with the remainder being additives such as plasticizers, fillers, colorants, pigments, UV-stabilizers, antioxidants, nucleating agents, and antibacterial additives [28]. The PE and weedmat mulches used in this study were both made of PE feedstock whereas the embossed BDM feedstock comprised of PLA (polylactic acid) and PBAT (polybutylene adipate-co-terephthalate). If ingredients differ in hydrophobicity or hydrophilicity, this could in turn influence the interaction between water droplets and resultant splash dispersal. In addition, different plastic mulches have varying surface characteristics due to their manufacturing (i.e., film vs. woven material vs. embossing), which could impact splash dispersal of pathogen spores and disease risk of fungal pathogens such as *B. cinerea.*

In conclusion, physical characteristics including permeability, surface physical features, and splashing dynamics of PE mulch, weedmat, and embossed BDM can impact splash dispersal dynamics of *B. cinerea*. Differences in the number of conidia dispersed among PE mulch, weedmat, and embossed BDM was observed. The embossed BDM facilitated greater dispersal of splashed conidia compared to PE mulch and weedmat. Based on these results, mulch physical features created by embossing may contribute to enhanced *B. cinerea* inoculum availability in strawberry production under plasticulture. However, the conidial concentrations observed among treatments were overall low and differences in splashed conidia may not be great enough to elicit differential responses in disease incidence at field scale. Future studies should focus on translating these differences into potential real-world risks within commercial strawberry fields for *B. cinerea* as well as other pathogens dispersed through rain splash.

## Supporting information

**S1 Table. Mulch tension measurements in a mulched 'Albion' strawberry field in Mount Vernon, WA.**
(DOCX)

**S2 Table. Uniformity tests for standardizing rain simulator system before running splash dispersal experiments.**
(DOCX)

**S3 Table. Assessment of conidial germination rates prior to running the splash dispersal Experiment.**
(DOCX)

**S4 Table. Germination rate of all dispersed *Botrytis cinerea* conidia recovered from the rain splash dispersal experiment.**
(DOCX)

**S1 Data. Data used in the experiment.**
(XLSX)

## Acknowledgments

We are thankful to Emma Rogers, Brenda Madrid, Rachel Schick, Dakota McFadden, and Roshani Baral for their help with application of the rain simulator system and data collection. We also would like to thank David DeVetter for input on rain simulator design.

## Author Contributions

**Conceptualization:** Lydia Tymon, Sunghwan Jung, Lisa Wasko DeVetter.

**Data curation:** Xuechun Wang, Hang Liu, Lisa Wasko DeVetter.

**Formal analysis:** Xuechun Wang, Chakradhar Mattupalli, Gary Chastagner, Sunghwan Jung, Lisa Wasko DeVetter.

**Funding acquisition:** Lydia Tymon, Sunghwan Jung, Lisa Wasko DeVetter.

**Investigation:** Xuechun Wang, Lydia Tymon, Zixuan Wu, Sunghwan Jung, Hang Liu, Lisa Wasko DeVetter.

**Methodology:** Xuechun Wang, Gary Chastagner, Sunghwan Jung, Lisa Wasko DeVetter.

**Project administration:** Lisa Wasko DeVetter.

**Resources:** Lisa Wasko DeVetter.

**Supervision:** Sunghwan Jung, Lisa Wasko DeVetter.

**Visualization:** Xuechun Wang, Sunghwan Jung, Hang Liu.

**Writing – original draft:** Xuechun Wang, Sunghwan Jung, Lisa Wasko DeVetter.

**Writing – review & editing:** Xuechun Wang, Chakradhar Mattupalli, Gary Chastagner, Lydia Tymon, Sunghwan Jung, Lisa Wasko DeVetter.

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
