## [Decision Letter · Decision Letter 0]

12 Dec 2022

PONE-D-22-30988Physical characteristics of soil-biodegradable and nonbiodegradable plastic mulches impact conidial splash dispersal of Botrytis cinereaPLOS ONE

Dear Dr. DeVetter,

Thank you for submitting your manuscript to PLOS ONE. After careful consideration, we feel that it has merit but does not fully meet PLOS ONE’s publication criteria as it currently stands. Therefore, we invite you to submit a revised version of the manuscript that addresses the points raised during the review process.

Reviewer’s comments on the manuscript have now been received. Having considered these comments alongside your paper there is need of major revisions.

The authors claimed the authors explore the impact of diverse mulches in *B. cinerea* conidia dispersion by water splashed. The manuscript is written in a concise manner but still need improvements suggested by the reviewers. A few of the suggestion are;

The experiments appear to have been done with rigor, using robust replicates, with no uncontrolled variables.

It is not clear to me the usage of the absorbent mat in the experiment. At what point is it added?

Also not clear at which point the conidia counting was performed. Was it after the 18hours of incubation? How long where the conidia at 1ºC?

Discussion should be improved as there is a need to correlate the results of current study with the previous findings. Also draw a putative applicable conclusion based on the regression for strawberry farms in their choices of mulches and/or plant distancing.

Furthermore the manuscript needs to be checked for grammatical errors and typo mistakes. English language should be improved by a native English speaker.

The detailed reviewer’s suggestions and revisions are attached that should be considered before finalizing the manuscript.

We look forward to receiving your revised manuscript.

Kind regards,

Raees Ahmed, Ph.D.

Academic Editor

PLOS ONE

Journal Requirements:

"This project was funded by the Washington State Department of Agriculture. We are thankful to Emma Rogers, Brenda Madrid, Rachel Schick, Dakota McFadden, and Roshani Baral for their help with application of the rain simulator system and data collection. We also would like to thank David DeVetter for input on rain simulator design. "

"This project was funded by the Washington State Department of Agriculture Specialty Crop Block Grant program (#K2863). Corresponding author, L.W.D., received the award. The funders had no role in study design, data collection and analysis, decision to publish, or preparation of the manuscript.

Website: https://agr.wa.gov/services/grant-opportunities/specialty-crop-block-grant-program "

5. We note that Figures 1 and 2 in your submission contain copyrighted images. All PLOS content is published under the Creative Commons Attribution License (CC BY 4.0), which means that the manuscript, images, and Supporting Information files will be freely available online, and any third party is permitted to access, download, copy, distribute, and use these materials in any way, even commercially, with proper attribution. For more information, see our copyright guidelines: http://journals.plos.org/plosone/s/licenses-and-copyright.

a. You may seek permission from the original copyright holder of Figures 1 and 2 to publish the content specifically under the CC BY 4.0 license. 

Additional Editor Comments:

Reviewer’s comments on the manuscript have now been received. Having considered these comments alongside your paper there is need of major revisions.

The authors claimed the authors explore the impact of diverse mulches in B. cinerea conidia dispersion by water splashed. The manuscript is written in a concise manner but still need improvements suggested by the reviewers. A few of the suggestion are;

The experiments appear to have been done with rigor, using robust replicates, with no uncontrolled variables.

It is not clear to me the usage of the absorbent mat in the experiment. At what point is it added?

Also not clear at which point the conidia counting was performed. Was it after the 18hours of incubation? How long where the conidia at 1ºC?

Discussion should be improved as there is a need to correlate the results of current study with the previous findings. Also draw a putative applicable conclusion based on the regression for strawberry farms in their choices of mulches and/or plant distancing.

Furthermore the manuscript needs to be checked for grammatical errors and typo mistakes. English language should be improved by a native English speaker.

The detailed reviewer’s suggestions and revisions are attached that should be considered before finalizing the manuscript.

Reviewers' comments:

Reviewer's Responses to Questions

**Comments to the Author**

1. Is the manuscript technically sound, and do the data support the conclusions?

Reviewer #1: Yes

Reviewer #2: Partly

Reviewer #3: Yes

2. Has the statistical analysis been performed appropriately and rigorously? 

Reviewer #1: Yes

Reviewer #2: Yes

Reviewer #3: Yes

3. Have the authors made all data underlying the findings in their manuscript fully available?

Reviewer #1: Yes

Reviewer #2: No

Reviewer #3: Yes

4. Is the manuscript presented in an intelligible fashion and written in standard English?

Reviewer #1: Yes

Reviewer #2: Yes

Reviewer #3: Yes

5. Review Comments to the Author

Reviewer #1: COMMENTS FOR THE AUTHOR

The manuscript ‘Physical characteristics of soil-biodegradable and non-biodegradable plastic mulches impact conidial

splash dispersal of Botrytis cinerea’ reports an interesting work providing important data regarding investigation of

splash dispersal dynamics of Botrytis cinerea when exposed to various plastic mulch surfaces.

In addition, several points need to be improved, as follows:

Introduction:

1- The author mentioned that plastic mulches impact splash dispersal of Botrytis cinerea conidia, but there isn’t

much focus on this statement in introduction with regard to references. Please clarify more.

2- Did the author investigate impact of different plastic mulches with varying surface characteristics with regard

to their manufacturing and formulation? They may have different impact on splash dispersal of pathogen

spores and disease risk of fungal pathogens such as B. cinerea. This should be clearly elaborated to avoid

confusions.

Material and Methodology:

1- From where unweathered PE, weedmat, and embossed BDM were taken, which author used in this study?

Source should be clearly mentioned.

2- The author mentioned that for each mulch treatment, a large plastic Petri dish (14 cm in diameter) was filled

with greenhouse growing medium. Was the experiment conducted in-vitro? Where the petri dish was placed

later? What about other parameters including temperature?

3- The author mentioned that for conidial splash dispersal of B. cinerea, experiment was conducted in a fully

enclosed screen house. The ground of screenhouse was covered with woven landscape polyethylene fabric.

Mention source.

4- To prepare the conidial suspension, the isolate was cultured on full-strength potato dextrose agar. Was

pathogenicity of botrytis cinerea also conducted?

5- Statistical analysis: Which ANOVA was used? Please clarify technical replicates.

Results:

1- While analysis of investigating impact of plastic mulches on conidial splash dispersal of Botrytis cinerea, did

the author applied any control in experiment? Clarify.

2- The discussion must be more developed to better value the results obtained and should go beyond the content

of paper, highlighting promising evidence for future application.

3- Latin names must be reported in italics.

4- The English form should be carefully revised.

Reviewer #2: In the article “Physical characteristics of soil-biodegradable and nonbiodegradable plastic mulches impact conidial splash dispersal of Botrytis cinerea”, the authors explore the impact of diverse mulches in B. cinerea conidia dispersion by water splashed. Through the development of a rain simulator system, conidial splashes are evaluated in terms of horizontal distance, as well as their germination potential.

The article is written in an easily comprehensible English and it is pleasant to read. The abstract sums well the findings of the authors, however, the last sentence should not be an element present in the abstract.

The introduction is well written, summarizing the current knowledge and gap on research where the paper fits. It exposes the relevant information for both acquainted and new readers on the topic. Some notes:

- B. cinerea is currently defined as hemibiotrophic, and therefore the categorisation of necrotrophic as in line 54 should be revised.

(Recommended: Veloso J, van Kan JAL. Many Shades of Grey in Botrytis-Host Plant Interactions. Trends Plant Sci. 2018 Jul;23(7):613-622. doi: 10.1016/j.tplants.2018.03.016. Epub 2018 Apr 30. PMID: 29724660.)

- Not only ripe fruits are susceptible to B. cinerea as said in line 55, there has been reports of green fruits infected. Consider including the word "post-harvest" too in line 56.

(Recommended: Agudelo-Romero P, Erban A, Rego C, Carbonell-Bejerano P, Nascimento T, Sousa L, Martínez-Zapater JM, Kopka J, Fortes AM. Transcriptome and metabolome reprogramming in Vitis vinifera cv. Trincadeira berries upon infection with Botrytis cinerea. J Exp Bot. 2015 Apr;66(7):1769-85. doi: 10.1093/jxb/eru517. Epub 2015 Feb 11. PMID: 25675955; PMCID: PMC4669548.)

-Line 74 and 75, the "respectively" is not clear in which pathogen is related to which disease.

M&M are very detailed, and in general clear and possible reproduced by others if wanted. The experiments appear to have been done with rigor, using robust replicates, with barely no uncontrolled variables.

Some notes/questions:

-It is not clear to me the usage of the absorbent mat in the experiment. At what point is it added? Please clarify in text and/or improve figure 3. Also, was the mat used or considered for the uniformity test? If not, why?

-It's not clear at which point the conidia counting was performed. Was it after the 18hours of incubation? How long where the conidia at 1ºC? (line 209)

-Please provide or include in supplementary a picture to go with S3 Table of the conidia counting and germination with scale of proper germ tube length. (line 211)

-data analysis: Please clarify what you mean with "while TibyT, and GibyG data were in percentages, so they were subjected to a normal distribution." Which test did you use to verify normal distribution? line 239.

Results are well exposed, though not always presented in the easiest way to read. Discussion is rather descriptive, unfortunately. Please explore and compare the results with the current literature already referenced by the authors in the introduction. Also draw a putative applicable conclusion based on the regression for strawberry farms in their choices of mulches and/or plant distancing.

Some notes/questions:

- line 260. How much of the volume was lost in infiltration in weedmat, how did that affect the experiment, and how did you solve it? Did the weedmat filtrate the conidia, or were they drained with the water?

- line 278 makes this paper redundant and irrelevant if the authors start to introduce already "Ifs” at the beginning of the discussion. Please reconsider its writing. The article is clear on the feedstock ingredients of each mulch used. I think it's irrelevant to highlight so much how different feedstocks can affect the results, when 3 are already being compared. Re-write paragraph starting at 271 to showcase the problematic as a research gap and not as an obstacle.

(Example: So far, it has been unclear if the feedstock ingredients....etc)

-How do you explain such variation in weedmat in the first distance, compared to the others?

-Further explain how permeability may promote/limit conidia dispersion, and the dis/advantage of weedmat over the other two mulches.

-Data in Table 3 could gain more by being presented as plot(s). X-axis maybe either by mulch or distance, for example.

-Line 362. Please consider referring other fruits more similar to strawberry, such as grapes or tomatoes, instead of just lentils.

-The non-replicated splash dynamics study is too extensive for a non-concluding experiment. If not better integrated in the overall article, maybe should be considered for supplementary data instead.

Regarding references, most of the cited literature is old (<20 years). More recent and revised literature is available, especially in botrytis.

Ref 17 and 18 are the same.

Figures notes/questions:

FIGURE 1:

Despite being an example of B. cinerea, it would be more interesting to have a picture with already visible conidia formation instead of just apparent mycelium, as conidia are the problematic of the article.

FIGURE 2:

If known, add in the caption the type of plastic mulch used in the picture.

FIGURE 3:

Centre water drop is not clear that is the conidial suspension. Please correct. Refer in the caption the conidia concentration used too.

FIGURE 4:

Any reason to use a different method to photograph the weedmat? How does its surface compare to the other two at the same magnification?

Axis in figure 4 are unreadable. I also think plots would gain by having the same scale, all aligned at y=0, for an easier comparison between the three mulches.

S4 table: what is the purpose of the letter b in the germination rate of PE 28 cm?

Small notes:

-GCi is defined a second time at 321. It was already done at 214.

-TCi is defined a second time at 282. It was already done at 213.

-Overall, choose between the term conidia or spore and make it uniform across the manuscript.

Overall, the experiments address an interesting problematic of strawberry farms and indeed meet the objectives and sub-objectives defined in the introduction. However, data is not properly explored to its full potential, and discussion lacks hypothesis beyond the descriptive results. In the current state, this article is not up to standards and need improvements. If improvements are achieved, this paper has potential for a quality publication.

Reviewer #3: it is a good piece of scientific work performed by the authors. the data was presented in a good and scientific way.

However some modifications are needed in the manuscript.

Introduction needs to be improved by the addition of more relevant literature and make easy to understand for the non-native english speakers.

Materials methods section is good

results and discussion section needs more elaboration and comparison with literature.

conclusion section needs complete revision and clearly mentioning of results.

6. PLOS authors have the option to publish the peer review history of their article (what does this mean?). If published, this will include your full peer review and any attached files.

Reviewer #1: No

Reviewer #2: **Yes: **Helena Santos

Reviewer #3: No

---

## [Author Response · Author response to Decision Letter 0]

27 Jan 2023

Please see the attachment "PLOS ONE Response to Review_final.docx". Our response is also copied from the letter below. 

Dear Dr. Raees Ahmed:

Thank you for the consideration of our manuscript, “Physical characteristics of soil-biodegradable and nonbiodegradable plastic mulches impact conidial splash dispersal of Botrytis cinerea” (Manuscript ID: PONE-D-22-30988), for publication in PLOS ONE. My co-authors and I are grateful for the reviews that will help strengthen the quality of our manuscript. Please see below a point-by-point description of how the manuscript has been revised based on reviewer feedback with our responses in red font [in the attachment; red font is not visible herein]. Furthermore, all changes in the manuscript have been made using Track Changes in MS Word per the journal’s requirements. Mention of line numbers refers to lines in the clean version of our submitted manuscript. We hope the revisions are satisfactory and welcome the opportunity to contribute to your journal. 

Editor Comments:

The experiments appear to have been done with rigor, using robust replicates, with no uncontrolled variables. 

Response: We are pleased the level of rigor is evident in the manuscript. 

It is not clear to me the usage of the absorbent mat in the experiment. At what point is it added? 

Response: The absorbent mat was used to limit secondary splash from outside of our primary inoculum source. This explanation has been added to the manuscript (line 224). 

Also not clear at which point the conidia counting was performed. Was it after the 18 hours of incubation? How long where the conidia at 1ºC?

Response: Conidia were counted immediately after 18 hours of incubation at 22 ℃ with plates held at 1 ℃ after the incubation period to limit conidia growth for a maximum of 4 hours while conidia were being counted. This has been clarified in the manuscript (lines 249-251). 

Discussion should be improved as there is a need to correlate the results of current study with the previous findings. Also draw a putative applicable conclusion based on the regression for strawberry farms in their choices of mulches and/or plant distancing.

Response: The discussion has been largely re-written and expanded to relate our study with previous findings. Putative applications for strawberry farms based on mulch characteristics and plant spacing has also been added. 

Furthermore the manuscript needs to be checked for grammatical errors and typo mistakes. English language should be improved by a native English speaker. 

Response: The manuscript has been reviewed multiple times and any remaining errors and typos have been corrected. Also, multiple authors including the corresponding author are native English speakers with English as their first and primary language. They have all reviewed, revised, and approve of the manuscript. 

Reviewer #1

The manuscript ‘Physical characteristics of soil-biodegradable and non-biodegradable plastic mulches impact conidial splash dispersal of Botrytis cinerea’ reports an interesting work providing important data regarding investigation of splash dispersal dynamics of Botrytis cinerea when exposed to various plastic mulch surfaces.

In addition, several points need to be improved, as follows:

Introduction:

1- The author mentioned that plastic mulches impact splash dispersal of Botrytis cinerea conidia, but there isn’t much focus on this statement in introduction with regard to references. Please clarify more.

Response: Thank you for your suggestion. We have expanded the introduction and references therein on splash dispersal of fungal plant pathogens. However, studies involving plastic mulches are limited and we could not find any references on splash dispersal of Botrytis cinerea with different plastic mulches. To our knowledge, we are the first research team to publish a study on plastic mulch impacts on Botrytis conidia splash dispersal (see lines 94-108).

2- Did the author investigate impact of different plastic mulches with varying surface characteristics with regard to their manufacturing and formulation? They may have different impact on splash dispersal of pathogen spores and disease risk of fungal pathogens such as B. cinerea. This should be clearly elaborated to avoid confusions. 

Response: Formulation of mulches was not a factor included in the design of this study, nor was manufacturing. This study focused on physical features of mulches, such as embossing and woven plastic fibers. We have limited our conclusions to focus on physical features and also mentioned formulation and manufacturing processes could be a future area of study in the discussion (lines 433-446). 

Material and Methodology:

1- From where unweathered PE, weedmat, and embossed BDM were taken, which author used in this study? Source should be clearly mentioned.

Response: All mulch materials were taken from the same mulch roll we had in stock for another study (Wang et al.,2022; citation updated at line 117). Mulch manufacturers are listed in Table 1. 

2- The author mentioned that for each mulch treatment, a large plastic Petri dish (14 cm in diameter) was filled with greenhouse growing medium. Was the experiment conducted in-vitro? Where the petri dish was placed later? What about other parameters including temperature? 

Response: The purpose of using greenhouse growing media was to mirror field conditions (i.e., mulch over soil) by suspending our mulch treatments over the media and adjusting the tension of the mulch so that it matched what we have measured in the field. The inoculum suspension was placed on top of the mulch and had no contact with the media except for in the instances where it percolated through for our weedmat treatment. This experiment was not done in-vitro. The Petri dish was collected after each trial and the mulch was discarded. Petri dishes were replaced with a sterile Petri dish after each replicate. We did not control for temperature during the rain simulation as we do not expect temperature to influence splash dispersal dynamics when above freezing. 

3- The author mentioned that for conidial splash dispersal of B. cinerea, experiment was conducted in a fully enclosed screen house. The ground of screenhouse was covered with woven landscape polyethylene fabric. Mention source.

Response: The screenhouse manufacturer and dimensions are now provided (lines150). We unfortunately do not have the manufacturing information of the landscape fabric, but it is representative of typical landscape fabrics. 

4- To prepare the conidial suspension, the isolate was cultured on full-strength potato dextrose agar. Was pathogenicity of botrytis cinerea also conducted?

Response: We conducted pathogenicity tests to confirm pathogenicity and updated Figure 1 to include an image of a strawberry fruit infected by the isolate used in this study. 

5- Statistical analysis: Which ANOVA was used? Please clarify technical replicates.

Response: We specified that multivariate ANOVA was used to analyze data (line 282). Technical replicates have been specified in the manuscript (please see lines 134, 180, 236, 275). 

Results:

1- While analysis of investigating impact of plastic mulches on conidial splash dispersal of Botrytis cinerea, did the author applied any control in experiment? Clarify.

Response: We believe the reviewer is referring to experimental controls. In our case, standard PE is considered a control because it is the most widely used mulch material in commercial strawberry systems. Our alternative treatments in this study were BDM and weedmat, as mentioned in the introduction (lines 109-113). 

2- The discussion must be more developed to better value the results obtained and should go beyond the content of paper, highlighting promising evidence for future application.

Response: The discussion has been re-written and we have added further discussion on plant spacing and future areas of research. 

3- Latin names must be reported in italics.

Response: This was double checked and we confirm all Latin names are in italics. It is possible italics were removed when the manuscript was converted to a format for peer-review. 

4- The English form should be carefully revised.

Response: Multiple authors including the corresponding author are native English speakers with English as their first and primary language. They have all reviewed, revised, and approve of the manuscript including the English form. We believe the English is acceptable for an English-speaking/reading audience. We also respectfully add that one of the reviewers felt the manuscript was “written in an easily comprehensible English and it is pleasant to read.” Therefore, we feel the English form is acceptable. 

Reviewer #2

Reviewer #2: In the article “Physical characteristics of soil-biodegradable and nonbiodegradable plastic mulches impact conidial splash dispersal of Botrytis cinerea”, the authors explore the impact of diverse mulches in B. cinerea conidia dispersion by water splashed. Through the development of a rain simulator system, conidial splashes are evaluated in terms of horizontal distance, as well as their germination potential.

The article is written in an easily comprehensible English and it is pleasant to read. The abstract sums well the findings of the authors, however, the last sentence should not be an element present in the abstract. 

Response: Thank you for the comments. We have revised the last sentence of the abstract to, “However, differences in conidial concentrations observed among treatments were low and may not be pathologically relevant.”. We feel it is important to be transparent in the abstract that despite differences being observed, it’s still uncertain if differences manifest into differences in disease incidence. 

The introduction is well written, summarizing the current knowledge and gap on research where the paper fits. It exposes the relevant information for both acquainted and new readers on the topic. Some notes:

- B. cinerea is currently defined as hemibiotrophic, and therefore the categorisation of necrotrophic as in line 54 should be revised.(Recommended: Veloso J, van Kan JAL. Many Shades of Grey in Botrytis-Host Plant Interactions. Trends Plant Sci. 2018 Jul;23(7):613-622. doi: 10.1016/j.tplants.2018.03.016. Epub 2018 Apr 30. PMID: 29724660.)

Response: Thank you for the helpful information! This has been corrected and the reference was updated as well (line 72). 

- Not only ripe fruits are susceptible to B. cinerea as said in line 55, there has been reports of green fruits infected. Consider including the word "post-harvest" too in line 56.

(Recommended: Agudelo-Romero P, Erban A, Rego C, Carbonell-Bejerano P, Nascimento T, Sousa L, Martínez-Zapater JM, Kopka J, Fortes AM. Transcriptome and metabolome reprogramming in Vitis vinifera cv. Trincadeira berries upon infection with Botrytis cinerea. J Exp Bot. 2015 Apr;66(7):1769-85. doi: 10.1093/jxb/eru517. Epub 2015 Feb 11. PMID: 25675955; PMCID: PMC4669548.)

Response: This has been corrected and the reference has been added as well (see lines 73). 

-Line 74 and 75, the "respectively" is not clear in which pathogen is related to which disease.

Response: This has been clarified (line 96). 

M&M are very detailed, and in general clear and possible reproduced by others if wanted. The experiments appear to have been done with rigor, using robust replicates, with barely no uncontrolled variables.

Some notes/questions:

-It is not clear to me the usage of the absorbent mat in the experiment. At what point is it added? Please clarify in text and/or improve figure 3. Also, was the mat used or considered for the uniformity test? If not, why?

Response: We have clarified the role of the absorbent mat and when it was added (lines 223-226). Fig. 3 has been updated to provide further clarification. We did not use the mat for uniformity tests because our purpose of doing uniformity tests was to characterize and adjust the rain simulator system for maximum uniformity, so the same amount of water accumulated in each direction from the inoculum source. 

-It's not clear at which point the conidia counting was performed. Was it after the 18hours of incubation? How long where the conidia at 1ºC? (line 209)

Response: Conidia counting procedures after incubation at 22 °C and duration at 1ºC has been clarified (see lines 247-252). 

-Please provide or include in supplementary a picture to go with S3 Table of the conidia counting and germination with scale of proper germ tube length. (line 211)

Response: We appreciate the suggestion. Unfortunately, our lab microscope does not connect to a camera so we cannot provide images of conidia counting and germination with scale of proper germ tube length.

-data analysis: Please clarify what you mean with "while TibyT, and GibyG data were in percentages, so they were subjected to a normal distribution." Which test did you use to verify normal distribution? line 239.

Response: We consulted and confirmed our statistical analysis approach with the Washington State University Center for Interdisciplinary Statistical Education and Research (CISER) consulting team (https://gradschool.wsu.edu/pdi/ciser/). They advised for count data we used log transformations for analysis as needed but for data in percentages we directly use untransformed data to fit the normal distribution. We first checked the normal distribution by using Rstudio function hist (). Our statistical consultant did not advise normality tests. In fact, in a follow-up conversation our consultant recommended visualization approaches over tests for assessments of normality. Our approach to assess normality has been added to the manuscript (line 279-281). 

Results are well exposed, though not always presented in the easiest way to read. Discussion is rather descriptive, unfortunately. Please explore and compare the results with the current literature already referenced by the authors in the introduction. Also draw a putative applicable conclusion based on the regression for strawberry farms in their choices of mulches and/or plant distancing.

Some notes/questions:

- line 260. How much of the volume was lost in infiltration in weedmat, how did that affect the experiment, and how did you solve it? Did the weedmat filtrate the conidia, or were they drained with the water?

Response: Water loss through weedmat during rain simulation was variable among our five replicates. The volume of water loss in these five replicates was 9, 1, 5, 7, and 4 mL of a total volume of 10 mL over the course of 2-minutes. In addition, our permeability experiment shows that approximately 2 mL of a 10 mL solution of water infiltrated through our weedmat treatment in 2 minutes (Fig. 5). The conditions of our rain simulator led to more variability in water loss, and we attribute this to the action of the rain simulator increasing water loss through the interaction of our treatment upon which the suspension rested upon and the falling droplets. Water droplets falling from the sprinkler heads also could add to the volume of water in the silicone ring, adding to variation in water loss during the rain simulation. In contrast, there was no sprinkler system operating during initial permeability assessments that generated data used to create Fig. 5. We did not attempt to control for this variation in water loss as we wanted to mirror real-world conditions in which weedmat is used. The variability in water lost through infiltration may impact droplet size and the number of conidia dispersed via rain splash, but we are unable to quantify this given we did not measure nor determine if weedmat filtrated the conidia. 

- line 278 makes this paper redundant and irrelevant if the authors start to introduce already "Ifs” at the beginning of the discussion. Please reconsider its writing. The article is clear on the feedstock ingredients of each mulch used. I think it's irrelevant to highlight so much how different feedstocks can affect the results, when 3 are already being compared. Re-write paragraph starting at 271 to showcase the problematic as a research gap and not as an obstacle.

(Example: So far, it has been unclear if the feedstock ingredients....etc)

Response: Thank you for your advice. We re-wrote this paragraph (see lines 430-446). 

-How do you explain such variation in weedmat in the first distance, compared to the others?

Response: Although not definitive, the variation in weedmat could be due to the semi-permeability and/or the irregular pattern of splashing due to weedmat’s roughness. This is explained in lines 299-302 and 365-374.

-Further explain how permeability may promote/limit conidia dispersion, and the dis/advantage of weedmat over the other two mulches.

Response: Thank you for the advice. We have commented on this aspect (see lines 299-302). 

-Data in Table 3 could gain more by being presented as plot(s). X-axis maybe either by mulch or distance, for example.

Response: Thank you for the suggestion. We reflected on this same point during initial manuscript preparation and submission. We believe it is optimal to present these data in table format so we can clearly show the value of our standard errors. Therefore, we respectfully prefer to keep data in Table 3 in its present table format. 

-Line 362. Please consider referring other fruits more similar to strawberry, such as grapes or tomatoes, instead of just lentils.

Response: This has been expanded to include other fruits, including grape (see lines 392-400). 

-The non-replicated splash dynamics study is too extensive for a non-concluding experiment. If not better integrated in the overall article, maybe should be considered for supplementary data instead.

Response: We appreciate the comments. We have reduced the extensive nature by dividing this component of the manuscript into the materials and methods (lines 139-147 and 259-268) and results and discussion (lines 349-356 and 357-374) sections. We also clearly stated this component is non-replicated and presents preliminary observations. These observations are constructive in interpreting our findings, so we prefer to not place these data in the supplementary files. 

Regarding references, most of the cited literature is old (<20 years). More recent and revised literature is available, especially in botrytis.

Response: We added more recent literature about Botrytis. Regarding splash dispersal, we cited Dr. Larry Madden’s seminal research on splash dispersal of fungal spores as this work is foundational and essential to reference. Recent studies are very rare and less robust compared to his. 

Ref 17 and 18 are the same. 

Response: These references are not the same, but different chapters. We have reviewed and made sure the references are correct. 

Figures notes/questions:

FIGURE 1:

Despite being an example of B. cinerea, it would be more interesting to have a picture with already visible conidia formation instead of just apparent mycelium, as conidia are the problematic of the article.

Response: We have updated Fig. 1 but were not able to show visible conidia formation as we do not have a photo of this. Instead, we added a photo showing the pathogenicity of our isolate which shows sporulation. We would also like to keep the original photo to demonstrate what B. cinerea looks like in a field situation, which may be of interest for the diverse audience of PLOS ONE. We have clarified in the caption that Fig. 1B shows sporulating B. cinerea. 

FIGURE 2:

If known, add in the caption the type of plastic mulch used in the picture.

Response: We have added in the caption that the mulch is polyethylene (PE).

FIGURE 3:

Centre water drop is not clear that is the conidial suspension. Please correct. Refer in the caption the conidia concentration used too.

Response: We corrected and used a blue dot with three orange arrows pointing to the conidial suspension.

FIGURE 4:

Any reason to use a different method to photograph the weedmat? How does its surface compare to the other two at the same magnification?

Response: The reason is that weedmat is a woven fabric made of material strips. This is different from PE and BDM, which are melt-spun films. When observing under a microscope, photos of large magnifications used for PE and BDM do not show this special mulch structure. For PE and BDM, large magnifications are needed to observe the mulch surface structures. 

Axis in figure 4 are unreadable. I also think plots would gain by having the same scale, all aligned at y=0, for an easier comparison between the three mulches.

Response: We apologize the axes in Figure 4 got left off during image conversion during preparation for submission to PLOS ONE. Axes are now depicted in the resubmission. The scales have also been adjusted and are all the same. Thank you for your attention to this.

S4 table: what is the purpose of the letter b in the germination rate of PE 28 cm?

Response: Letter b is the footnote and this was clearly showed under the table. Thanks! 

Small notes:

-GCi is defined a second time at 321. It was already done at 214.

-TCi is defined a second time at 282. It was already done at 213.

Response: We have revised notation and definition of TCi and GCi.

-Overall, choose between the term conidia or spore and make it uniform across the manuscript.

Response: Thanks! We corrected and changed spores to conidia where appropriate. 

Overall, the experiments address an interesting problematic of strawberry farms and indeed meet the objectives and sub-objectives defined in the introduction. However, data is not properly explored to its full potential, and discussion lacks hypothesis beyond the descriptive results. In the current state, this article is not up to standards and need improvements. If improvements are achieved, this paper has potential for a quality publication.

Response: Thank you very much for your advice! We have expanded data exploration and our overall discussion and hope this improves the quality for publication. 

Reviewer #3

It is a good piece of scientific work performed by the authors. the data was presented in a good and scientific way. 

Response: We are appreciative that the reviewer recognizes the scientific work presented in the manuscript. 

However some modifications are needed in the manuscript. 

Response: Modifications have been made based on the feedback. 

Introduction needs to be improved by the addition of more relevant literature and make easy to understand for the non-native English speakers. 

Response: We appreciate the feedback. Please note this is an interdisciplinary paper that combines horticulture, plant pathology, material science, and physics. Therefore, there may be some terms a reader may be unfamiliar with and requires them to look up some terminology or other literature if it’s outside of their area of expertise. PLOS ONE publishes multidisciplinary and interdisciplinary studies, which is why we targeted the journal for this study. Given the interdisciplinary nature and aim of PLOS ONE, we do not believe the text should be adjusted for non-native English speakers. Furthermore, the journal is published in English so we respectfully submit that discipline-specific terms should be used and presented in customary English form. We did expand the introduction and included relevant literature. 

Materials methods section is good 

Response: We are glad the methods section appears complete to the reviewer. Changes have been made based on input from other reviewers. 

results and discussion section needs more elaboration and comparison with literature. 

Response: We have expanded the discussion to include more comparisons with previous literature. However, please note published literature on the topic of pathogen splash dispersal as a function of groundcover (i.e., mulch) is very limited and this lack of research necessarily limits the scope of our discussion. 

conclusion section needs complete revision and clearly mentioning of results. 

Response: The conclusion has been revised with the results clearly articulated. 

Journal Requirements:

and 

"This project was funded by the Washington State Department of Agriculture. We are thankful to Emma Rogers, Brenda Madrid, Rachel Schick, Dakota McFadden, and Roshani Baral for their help with application of the rain simulator system and data collection. We also would like to thank David DeVetter for input on rain simulator design. "

"This project was funded by the Washington State Department of Agriculture Specialty Crop Block Grant program (#K2863). Corresponding author, L.W.D., received the award. The funders had no role in study design, data collection and analysis, decision to publish, or preparation of the manuscript.

Website: https://agr.wa.gov/services/grant-opportunities/specialty-crop-block-grant-program "

Response: We have removed the funding statement in the acknowledgements section of the manuscript. Our funding statement has been revised and is listed below:

“This project was funded by the Washington State Department of Agriculture Specialty Crop Block Grant program (#K2863). Additionally, this work was supported by Specialty Crops Research Initiative Award 2022-51181-38325 from the USDA National Institute of Food and Agriculture. Any opinions, findings, conclusions, or recommendations expressed in this publication are those of the author(s) and do not necessarily reflect the view of the U.S. Department of Agriculture. Corresponding author, L.W.D., received the awards. The funders had no role in study design, data collection and analysis, decision to publish, or preparation of the manuscript.

Website for WSDA: https://agr.wa.gov/services/grant-opportunities/specialty-crop-block-grant-program

Website for USDA SCRI: https://www.nifa.usda.gov/grants/funding-opportunities/specialty-crop-research-initiative "

Response: We have provided the study’s minimal underlying data set as Supporting Information files in this re-submission. 

5. We note that Figures 1 and 2 in your submission contain copyrighted images. All PLOS content is published under the Creative Commons Attribution License (CC BY 4.0), which means that the manuscript, images, and Supporting Information files will be freely available online, and any third party is permitted to access, download, copy, distribute, and use these materials in any way, even commercially, with proper attribution. For more information, see our copyright guidelines: http://journals.plos.org/plosone/s/licenses-and-copyright.

a. You may seek permission from the original copyright holder of Figures 1 and 2 to publish the content specifically under the CC BY 4.0 license. 

Response: Figures 1A and 2 were taken by corresponding author DeVetter. Figure 1B was taken by co-author Mattupalli’s graduate student, Roshani Baral. Images in Figures 1 and 2 are not copyrighted. They are available for any third party to access, download, copy, distribute, and use in any way. We have credited DeVetter and Baral for their photo contributions.

---

## [Decision Letter · Decision Letter 1]

17 Apr 2023

Physical characteristics of soil-biodegradable and nonbiodegradable plastic mulches impact conidial splash dispersal of Botrytis cinerea

PONE-D-22-30988R1

Dear Dr. DeVetter,

We’re pleased to inform you that your manuscript has been judged scientifically suitable for publication and will be formally accepted for publication once it meets all outstanding technical requirements.

Kind regards,

Abhay K. Pandey

Academic Editor

PLOS ONE

Additional Editor Comments (optional):

authors addressed all comments.

Reviewers' comments:

Reviewer's Responses to Questions

**Comments to the Author**

1. If the authors have adequately addressed your comments raised in a previous round of review and you feel that this manuscript is now acceptable for publication, you may indicate that here to bypass the “Comments to the Author” section, enter your conflict of interest statement in the “Confidential to Editor” section, and submit your "Accept" recommendation.

Reviewer #1: All comments have been addressed

Reviewer #2: All comments have been addressed

2. Is the manuscript technically sound, and do the data support the conclusions?

Reviewer #1: Yes

Reviewer #2: Yes

3. Has the statistical analysis been performed appropriately and rigorously? 

Reviewer #1: Yes

Reviewer #2: Yes

4. Have the authors made all data underlying the findings in their manuscript fully available?

Reviewer #1: Yes

Reviewer #2: Yes

5. Is the manuscript presented in an intelligible fashion and written in standard English?

Reviewer #1: Yes

Reviewer #2: Yes

6. Review Comments to the Author

Reviewer #1: Author has incorporated all suggestions carefully. Discussion is also improved to some extent. I am much satisfied

Reviewer #2: In the improved article “Physical characteristics of soil-biodegradable and nonbiodegradable plastic mulches impact conidial splash dispersal of Botrytis cinerea”, the authors explore the impact of diverse mulches in B. cinerea conidia dispersion by water splashed.

The authors addressed all my points and questions, defended their choices very well, and performed adequate modifications to the manuscript such as improvement of the discussion, correction of the figures, and clarification of the methods. I have no further comments to improve the manuscript, and I am rather pleased with the current version. I believe it fits the standards of PLOS ONE for publication and should be accepted. The authors should be proud of their work.

7. PLOS authors have the option to publish the peer review history of their article (what does this mean?). If published, this will include your full peer review and any attached files.

Reviewer #1: **Yes: **Dr. Gulshan Irshad Associate Professor

Reviewer #2: **Yes: **Helena Santos

---

## [Editor Report · Acceptance letter]

28 Apr 2023

PONE-D-22-30988R1 

Physical characteristics of soil-biodegradable and nonbiodegradable plastic mulches impact conidial splash dispersal of *Botrytis cinerea*

Dear Dr. DeVetter:

I'm pleased to inform you that your manuscript has been deemed suitable for publication in PLOS ONE. Congratulations! Your manuscript is now with our production department. 

Kind regards, 

on behalf of

Dr. Abhay K. Pandey 

Academic Editor

PLOS ONE